# Untargeted Microbial Exometabolomics and Metabolomics Analysis of *Helicobacter pylori* J99 and *jhp0106* Mutant

**DOI:** 10.3390/metabo11120808

**Published:** 2021-11-28

**Authors:** Cheng-Yen Kao, Pei-Yun Kuo, Hsiao-Wei Liao

**Affiliations:** 1Institute of Microbiology and Immunology, School of Life Science, National Yang Ming Chiao Tung University, Taipei 122, Taiwan; kaocy@ym.edu.tw (C.-Y.K.); togebi87303@gmail.com (P.-Y.K.); 2Department of Pharmacy, National Yang Ming Chiao Tung University, Taipei 122, Taiwan

**Keywords:** microbial metabolomics, exometabolomics, liquid chromatography-mass spectrometry, *Helicobacter pylori*, flagellum, *jhp0106* mutant

## Abstract

Untargeted metabolomic profiling provides the opportunity to comprehensively explore metabolites of interest. Herein, we investigated the metabolic pathways associated with Jhp0106, a glycosyltransferase enzyme in *Helicobacter pylori*. Through untargeted exometabolomic and metabolomic profiling, we identified 9 and 10 features with significant differences in the culture media and pellets of the wild-type (WT) J99 and *jhp0106* mutant (Δ*jhp0106*). After tentative identification, several phosphatidylethanolamines (PEs) were identified in the culture medium, the levels of which were significantly higher in WT J99 than in Δ*jhp0106*. Moreover, the reduced lysophosphatidic acid absorption from the culture medium and the reduced intrinsic diacylglycerol levels observed in Δ*jhp0106* indicate the possibility of reduced PE synthesis in Δ*jhp0106*. The results suggest an association of the PE synthesis pathway with flagellar formation in *H. pylori*. Further investigations should be conducted to confirm this finding and the roles of the PE synthesis pathway in flagellar formation. This study successfully demonstrates the feasibility of the proposed extraction procedure and untargeted exometabolomic and metabolomic profiling strategies for microbial metabolomics. They may also extend our understanding of metabolic pathways associated with flagellar formation in *H. pylori*.

## 1. Introduction

Metabolomics is a versatile investigation strategy that has been widely applied to accomplish various research aims, such as (in biological studies) determining the function of unknown genes and discovering biomarkers for clinical applications [1,2]. Exometabolomics (also known as metabolic footprinting) is a branch of metabolomics and is mainly focused on studying extracellular metabolite alteration. Both metabolomics and exometabolomics have been successfully performed by using mass spectrometry (MS) systems [3,4]. Untargeted metabolic profiling involves the comprehensive screening of all metabolites in an assayed sample, presenting a wide range of metabolites with diverse physicochemical properties. For example, the integration of electrospray ionization quadrupole time-of-flight (ESI-Q-TOF) MS with online databases enabled the detection of thousands of metabolic features in biological samples from *Escherichia coli* and human serum [5]. MS-based technologies provide sensitive and accurate metabolite measurements. However, the accurate identification and quantification of detected metabolic features remain challenging issues and are constrained by the availability of reference standards.

Microbial metabolomics is an essential component of cellular metabolomics, which involves the use of isolated or cultured microbial samples for various research purposes, such as investigations of bacterial metabolic pathways, understanding the cellular responses to microenvironmental changes, and exploring the relationships between microbes and diseases [6,7,8]. Microbial metabolomics is conducive to elucidating bacterial gene functions—the interest in which is growing—and their phenotypes [9]. For accurate and unbiased metabolomic profiling, the efficient preparation of microbial samples before analytical measurements is essential because it substantially influences the integrity of microbial metabolites. For example, optimizing quenching processes and extraction protocols is necessary for acquiring a sufficient amount of information on metabolites, and simple sample preparation approaches with fewer steps often yield higher precision in metabolite measurement [10]. Quenching processes and metabolite extraction procedures constitute two crucial parameters that affect the quality of microbial metabolomic data. Quenching by low-temperature treatment, before or after cell harvest or after cell washing, has been widely performed [11]. In this regard, cold methanol quenching is an efficient procedure [12]. However, this process is susceptible to metabolite leakage from cells. Considerable efforts have been devoted to determining the best method for mitigating or resolving this occurrence and its associated issues [13,14]. Protocols for metabolite extraction vary widely depending on the physicochemical properties of target metabolites. Cold methanol extraction is reportedly an efficient and reproducible methodology for metabolite extraction [15]. The applied extraction method is also associated with the results of metabolomic profiling with regard to matrix effects and the completeness of metabolite extraction. In essence, quenching-process efficiency, metabolite coverage, extraction completeness, and matrix effects should be considered in microbial sample preparation.

*Helicobacter pylori*, the dominant bacterial species in the human microbiota, is highly adapted for colonization in the human stomach [16]. *H. pylori* is believed to be present in approximately half of the human population and is associated with an increased risk of various diseases, including noncardiac gastric adenocarcinoma, gastric lymphoma, and peptic ulcers [17]. The flagellum of *H. pylori* is regarded as a crucial virulence factor that is vital to the colonization of gastric mucosa. Specifically, flagella-mediated motility is a pivotal factor for the virulence of *H. pylori*, which is associated with gastrointestinal diseases. Flagellin is the main component of the flagellum. Jhp0106, which was recently identified as a novel glycosyltransferase for flagellin glycosylation with pseudaminic acid, is essential for flagellar formation and consequently bacterial motility [18]. A better understanding of the metabolic pathways associated with Jhp0106 in flagellar formation would not only expand our knowledge of flagellin glycosylation and formation in clinically relevant pathogens but also benefit the development of innovative nonantibiotic treatments.

Relatively few studies have involved the metabolomic profiling of *H. pylori* isolates; the literature mainly focuses on biofilm formation and metabolic host–bacterium interactions [19,20]. This study investigates the metabolic pathways related to the flagellar formation enzyme Jhp0106 by employing exometabolomic and metabolomic strategies. *H. pylori* with the *jhp0106* mutation have previously been reported with no flagellar formation [18]. Through the untargeted exometabolomic profiling, we discovered the metabolite uptake and secretion associated with Δ*jhp0106*. While the untargeted metabolic profiling looked for the intrinsic metabolic pathway alteration associated with Δ*jhp0106*. After evaluation of the metabolite extraction method, microbial samples were extracted and applied for microbial exometabolomics and metabolomics, such that the metabolite alterations associated with flagellar formation could be examined.

## 2. Results and Discussion

### 2.1. Optimization of the Extraction Solvent Volume for the Culture Media and Bacterial Cell Pellets

This was a pilot study for finding potential metabolic pathways to guide research in submetabolomics or targeted metabolomics. Considering the favorable quenching efficiency and metabolite coverage provided by methanol, we decided to simultaneously quench and extract metabolites with methanol after a cell-washing protocol for subsequent untargeted metabolomic analysis.

Incomplete extraction and matrix effects may affect metabolite abundance. Therefore, a higher solvent extraction volume would be conducive to improving extraction efficiency and mitigating the impacts of the matrix effect. Thus, we first performed two continuous extractions to examine the metabolite residue in the precipitated protein residue. Next, 100-μL aliquots of culture media were extracted using 400 μL of methanol, and 3124 features with a signal-to-noise ratio of >3 and without signal saturation (intensity > 2 × 10^6^) were identified through untargeted metabolomic profiling. The bacteria pellets obtained from the 1.0-mL bacterial suspensions were extracted using 1.5 mL of methanol, and 2228 features with a signal-to-noise ratio of >3 and without signal saturation were observed. The number of detected features in both the extracted culture media and the bacterial pellets was acceptable for the subsequent untargeted metabolomic analysis. We next selected some of the observed metabolites (Appendix A) with different physicochemical properties in the culture media and bacterial pellets as tools for determining the completeness of the proposed extraction procedure. The matrix effect was corrected by using SIL-ISs or ISs, considering the compounds’ natures or the similarity of retention times (Appendix A). To evaluate the metabolite residue after methanol extraction, we performed continuous extraction tests. Briefly, 100 μL of culture media was continuously extracted using 400 μL of methanol, and *H. pylori* pellets were continuously extracted by using 1500 μL of methanol. The observed metabolite signals in the second extraction were normalized to the metabolite signals in the first extraction. As shown in Figure 1, the selected metabolites all exhibited less than 12.5% normalized abundance in the second extraction compared with in the first extraction. Even with the consideration of possible analytical errors, these results indicate an acceptable extraction efficiency for subsequent untargeted exometabolomic and metabolomic profiling of the culture media and bacterial pellets.

### 2.2. Untargeted Exometabolomic Profiling of Culture Media from H. pylori

Investigation of the metabolite alteration in the culture media facilitates the understanding of metabolite uptake or release from *H. pylori* during cultivation (*n* = 3 batch cultivation). The untargeted exometabolomic profiling of metabolites in the culture media obtained from WT J99 and Δ*jhp0106 H. pylori* was performed (three technical replicates for each sample). We first identified 3275 features by using MS-DIAL, and these features were subjected to PLS−DA (Figure 2a) to determine the features with higher variable importance in projection (VIP) scores, and heat maps were generated to visualize the distribution of these features in WT J99 and Δ*jhp0106* (Figure 2b). Additionally, we also conducted the two-sample *t*-test for the obtained features with a false discovery rate (FDR) adjusted *p*-value cut-off of 0.01 (Appendix A). We further evaluated these features by a volcano plot, and finally we obtained 11 features with a significant between-group difference (Figure 2c). After the extracted ion chromatograms of these features were examined, two features were manually removed because they had relatively low signal-to-noise ratios. Boxplots of the nine obtained features are displayed in Figure 2d.

Appendix A presents the observed *m*/*z*, theoretical *m*/*z*, retention time, tentatively identified compound names and formulas, and mass errors (in ppm) of these features. Two were unknown compounds and were tentatively identified as peptides by METLIN. Three features possibly originated from lysophosphatidic acid with an 18:2 lysophosphatidic acid (LPA) fatty acid moiety, owing to their identical retention times. The remaining features were tentatively identified as cyclic PA (cPA) 16:0 and phosphatidylethanolamines (PEs) such as PE 31:1, PE 32:0, and PE 35:1. These identities were further confirmed according to retention times and adducts were observed to have formed. Moreover, the identifies were supported by the observation that the fragments obtained from the lipid standard mixture contained PE and LPA.

### 2.3. Untargeted Metabolomic Profiling of H. pylori Pellets

The untargeted metabolomic profiling of metabolites of WT J99 and Δ*jhp0106 H. pylori* was performed to investigate the intrinsic alteration of *jhp0106* mutation in *H. pylori* (three technical replicates for every sample from three batch cultivation). We first identified 2041 features from MS-DIAL. PLS−DA (Figure 3a) and heat map (Figure 3b) reveal the distribution of these features in WT J99 and Δ*jhp0106 H. pylori*., and a volcano plot (Figure 3c) shows 49 features demonstrating significant between-group differences. Additionally, we also performed the two-sample *t*-test for the obtained features with an FDR adjusted *p*-value cut-off of 0.01. (Appendix A) Among these obtained features, 39 were removed because they had low signal-to-noise ratios or exhibited higher levels in the culture media. The aim was to prevent the possible contamination of culture media residue. Boxplots of the 10 features obtained are displayed in Figure 3d.

To identify the formulas of the features, we used databases such as METLIN and the Human Metabolome Database. Appendix A presents the observed *m*/*z*, theoretical *m*/*z*, retention time, tentatively identified compound names and formulas, and mass errors (in ppm) of these features. Six of the features were identified as different adducts or fragments of diacylglycerol (DG) 32:0 and DG 34:0. One of the features was identified as an unknown compound with a tentative formula of C_50_H_82_O. The remaining features were tentatively identified as PE 37:1, triacylglycerol (TG) 49:8, and phosphorylethanolamine. However, the obtained features, including phosphorylethanolamine, may have originated from insource degradation. Further confirmation with the reference standard is necessary.

### 2.4. Application of Untargeted Exometabolomic and Metabolomic Profiling to Exploring Flagellar Formation–Associated Metabolites and Metabolic Pathways

Jhp0106 was previously identified as a flagellar formation protein [21]. However, Δ*jhp0106 H. pylori* has never been reported to display flagellar formation. To study the metabolite alteration associated with the *jhp0106* mutation, we compared the tentatively identified metabolites from the culture media and *H. pylori* pellets. We further compared the level of identified metabolites with control culture media blanks to determine metabolite uptake or secretion. The results are presented in Figure 4a.

The PE levels were significantly higher in WT J99 than in Δ*jhp0106 H. pylori*. PE is an essential lipid component of *H. pylori*, but the functions of PEs have yet to be fully elucidated. One study reported that PEs on cell membranes can function as steroid-binding lipids [22]. In the present study, we observed elevated PE levels in the culture media after *H. pylori* cultivation. Moreover, the WT J99 exhibited higher PE levels than did Δ*jhp0106 H. pylori* (Figure 2d). In addition, the concentration of phosphorylethanolamine in Δ*jhp0106 H. pylori* was lower than that in WT J99. These results suggest that PE synthesis is reduced in Δ*jhp0106 H. pylori*, giving rise to elevated phosphorylethanolamine levels and reduced PE levels in the culture media and in Δ*jhp0106 H. pylori*. Since PEs is the major lipid species on the cell membrane of *H. pylori*, it indicates possibly elevated exosomes in the culture media. A more detailed study focused on the metabolic profiling of the isolated exosomes is required to prove this hypothesis.

The reduced levels of LPA 18:2 and cPA 16:0 in the culture media indicate the uptake of LPA 18:2 and cPA 16:0 from culture media after the cultivation of WT J99. The culture media from Δ*jhp0106 H. pylori* did not differ significantly from the control culture media blanks (Figure 2d). Although WT J99 uptake more LPA 18:2 and cPA 16:0 compared to Δ*jhp0106 H. pylori* from medium, LPA 18:2 and cPA 16:0 did not differ significantly inside the WT J99 bacteria cell. From these results, we can infer that LPA and cPA may be quickly synthesized or metabolized into other metabolites in WT J99.

The levels of DG 32:0 and DG 34:0 were lower in Δ*jhp0106 H. pylori* than in WT J99, whereas the levels of TG 49:8 were higher in Δ*jhp0106 H. pylori* than in WT J99 (Figure 3d). These results indicate the possible reduced synthesis of DG from TG in Δ*jhp0106 H. pylori*. 

In summary, we postulate that the observed metabolite alterations may be related to PE synthesis, as presented in Figure 4b. The reduction in PE synthesis may in turn lead to the reduced upstream synthesis of DG from TG as well as from the uptake of LPA and cPA. As mentioned, Jhp0106 constitutes an essential enzyme for flagellar formation. The present findings suggest a link between reduced PE synthesis and flagellar formation. Submetabolomic analysis focused on PEs or exosome-targeted subsample analysis are possible directions for confirming our findings and further exploring the roles and functions of PEs.

## 3. Materials and Methods

### 3.1. Chemicals and Materials

Formic acid (99%), ammonium acetate, and amino acid standards were obtained from Sigma-Aldrich Co. (St. Louis, MO, USA). High-performance liquid chromatography (HPLC)-grade methanol (MeOH) was purchased from Scharlau Chemie (Sentmenat, Barcelona, Spain). Isopropyl alcohol was supplied by J.T. Baker (Phillipsburg, NJ, USA). Stable isotope-labeled amino acid mix solution 1 was obtained from Supelco (Bellefonte, PA, USA). LightSPLASH and Cer/Sph Mixture I were obtained from Avanti Polar Lipids (Alabaster, AL, USA).

### 3.2. UHLPC-ESI-Q-TOF-MS System

Untargeted metabolomic profiling of plasma samples was conducted using an Agilent 1290 ultra-high-performance liquid chromatography (UHPLC) system (Agilent Technologies, Waldbronn, Germany) and a Bruker maXis ultra-high-resolution (UHR) TOF mass spectrometer (Bruker Daltonics, Bremen, Germany). A Thermo Scientific Accucore aQ C18 Polar Endcapped LC column (2.1 mm × 100 mm, 1.9 μm) was used to separate the polar metabolites from the nonpolar metabolites. Mobile phase A consisted of 10-mM ammonium acetate and 0.1% formic acid in deionized water. Mobile phase B consisted of 10-mM ammonium acetate and 0.1% formic acid in a 2:3 mixture of methanol and isopropanol. The gradient elution of 0.4 mL min^−1^ was applied under the following conditions: 0 to 2 min, 0% mobile phase B; 2–3 min, 0–5% mobile phase B; 3–5 min, 5–99% mobile phase B; and 5–15 min, 99% mobile phase B. This was followed by 2 min of column re-equilibration with 0% mobile phase B. The sample injection volume was 10 μL. The autosamplers and column oven were maintained at temperatures of 4 and 40 °C, respectively. Regarding ESI, the positive mode was applied with a dry gas temperature of 200 °C, dry gas flow rate of 8 L min^−1^, nebulizer gas pressure of 2 bar, capillary voltage of 4500 V, and end plate offset potential of 500 V. The mass spectra were acquired in full-scan mode in the range of 50–1500 *m/z*.

### 3.3. Bacterial Cultivation

The wild-type (WT) J99 and *jhp0106* mutant (Δ*jhp0106*) used in this study are described in our previous study [21]. In brief, *H. pylori* was grown on CDC anaerobic blood agar (BBL, Microbiology Systems, Cockeysville, MD, USA) or in Brucella broth containing 10% (*v*/*v*) horse serum (Gibco BRL, Life Technologies, Rockville, MD, USA) at 37 °C in microaerophilic conditions (5% O_2_, 10% CO_2_, and 85% N_2_). To perform metabolic assays, *H. pylori* cells were grown on Brucella/10% horse serum agar plates for 32–48 h, transferred to a 50-mL mixture of Brucella broth containing 10% horse serum at an initial optical density, at 600 nm (OD_600_), of 0.2, and inoculated at 37 °C with 18 h of shaking (150 rpm) to reach the mid-log phase.

### 3.4. Sample Preparation: Culture Media and Bacterial Cell Pellets

Bacterial counts must be adjusted before sample preparation, and the corresponding volume of extraction solvent should be optimized. To ensure homogeneity in bacterial counts between samples, we adjusted the bacterial suspensions according to their OD_600_; specifically, the suspensions were adjusted to an OD_600_ of 1.20 ± 0.05. This was achieved by adjusting the volume of the culture media before the quenching and extraction processes. After adjustment, 1-mL aliquots of bacterial suspensions were centrifuged and washed twice with 1 mL of PBS.

The bacterial cell pellets were obtained by centrifuging 1-mL aliquots of bacterial suspensions at 3000× *g* for 5 min. Medium residue was removed by washing the bacterial cell pellets in 1.0 mL of phosphate-buffered saline twice. Subsequently, the reconstituted bacterial cell pellets were quenched and extracted by using 1.5 mL of cold methanol. The deproteinized sample was centrifuged at 15,000× *g* for 5 min, and the supernatant was then passed through a 0.22-μm regenerated cellulose membrane syringe filter (RC-4, Sartorius, Göttingen, Germany) and stored at −20 °C until UHLPC-ESI-Q-TOF-MS analysis. To compensate for the matrix effects, the mixed stable-isotope-labeled (SIL) internal standards (ISs) were spiked at concentrations of 12 to 25 μM, and the ISs were spiked at the concentration of 1 μg mL^−1^ after metabolite extraction.

Subsequently, 100 μL of culture media was protein precipitated and extracted by using 400 μL of methanol. The deproteinized sample was centrifuged at 15,000× *g* for 5 min, after which the supernatant was passed through the same 0.22-μm membrane syringe filter before UHLPC-ESI-Q-TOF-MS analysis.

We performed three biological replicates, which means three different batches of WT and MT *H. pylori* were collected. Each batch collected one sample for both WT and MT *H. pylori*. Three technical repeats were performed for all collected samples.

### 3.5. Data Analysis

All the data obtained from the maXis UHR TOF mass spectrometer were processed using MS-DIAL software (http://prime.psc.riken.jp/, accessed 28 November 2021) and Bruker Compass DataAnalysis software v4.1. To identify the features with significant differences, an intensity cutoff of 3000 and a retention time alignment of 0.2 min were set for MS-DIAL. Next, these features were subjected to PLS−DA to determine the features with higher variable importance in projection (VIP) scores, and heat maps were generated to visualize the distribution of these features. The volcano plot was used to present features with a fold change of >2 and a *p* value of <0.05. Additionally, we also conducted the two-sample *t*-test for the obtained features with an FDR-adjusted *p*-value cut-off of 0.01. The acquired features with interest were identified by searching the METLIN Metabolite and Chemical Entity Database [23] (METLIN hereafter) and the Human Metabolome Database [24]. The identification of all the features in this study was level 3 identification by matching precursor *m*/*z* with the Metlin database. Partial least squares discriminant analysis (PLS−DA) was performed, and volcano plots were generated using the web-based metabolomics data processing tool MetaboAnalyst 5.0 (https://www.metaboanalyst.ca/, accessed 28 November 2021). For normalization, autoscaling mode was applied (mean-centered and divided by the standard deviation of each metabolite). The detailed data analysis workflow was shown in Appendix A.

## 4. Conclusions

Microbial metabolomics is receiving increasing attention, with scholarly efforts ranging from exploring the relationships between bacteria and disease by examining microbiota to identifying potentially innovative treatments by investigating metabolic pathways in bacteria. To perform an untargeted metabolomic profiling study, proper metabolite extraction is critical to providing comprehensive metabolite coverage and ensuring completed metabolite extraction. As this was a pilot study for identifying possible metabolite alterations associated with flagellar formation in *H. pylori*, we selected cold methanol as the extraction solvent because of its favorable quenching efficiency and coverage of the extracted metabolites. Extraction completeness was verified through consecutive extraction of 1.0-mL aliquots of *H. pylori* suspension with an approximate OD_600_ of 1.2. These results demonstrate that metabolites can be detected in the second extraction in limited amounts and that the extraction process was sufficiently thorough. 

To our knowledge, this is the first study to investigate the possible metabolite alterations associated with the flagellar formation of *H. pylori*. In this study, exometabolomic and metabolomic profiling were employed to study metabolite changes related to flagellar formation. We identified 9 and 10 metabolites that differed significantly between WT J99 and Δ*jhp0106 H. pylori* from the culture media and the *H. pylori* cell bodies, respectively. These tentatively identified metabolites indicate the possibility that the PE synthesis pathway is associated with flagellar formation in *H. pylori*. Further in-depth studies are warranted to confirm the potential roles and functions of PE in flagellar formation; these may include PE-centered submetabolomic analysis or exosome-targeted subsample analysis. Our demonstrated extraction process is expected to benefit future research on untargeted microbial exometabolomics and metabolomics.

## Figures and Tables

**Figure 1 metabolites-11-00808-f001:**
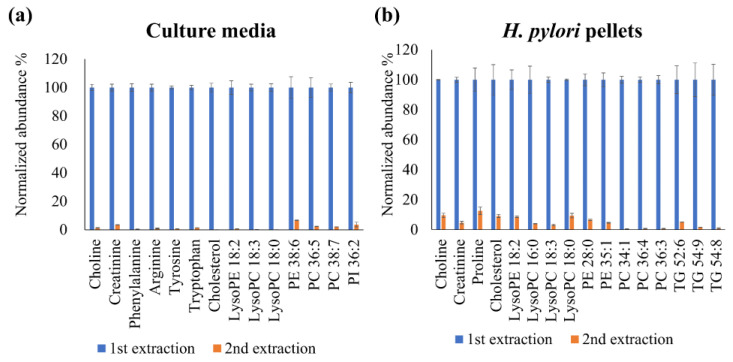
Continuous extraction tests of culture media (100 μL), performed with 400 μL of methanol (**a**), and of *H. pylori* pellets, performed with 1500 μL of methanol (**b**).

**Figure 2 metabolites-11-00808-f002:**
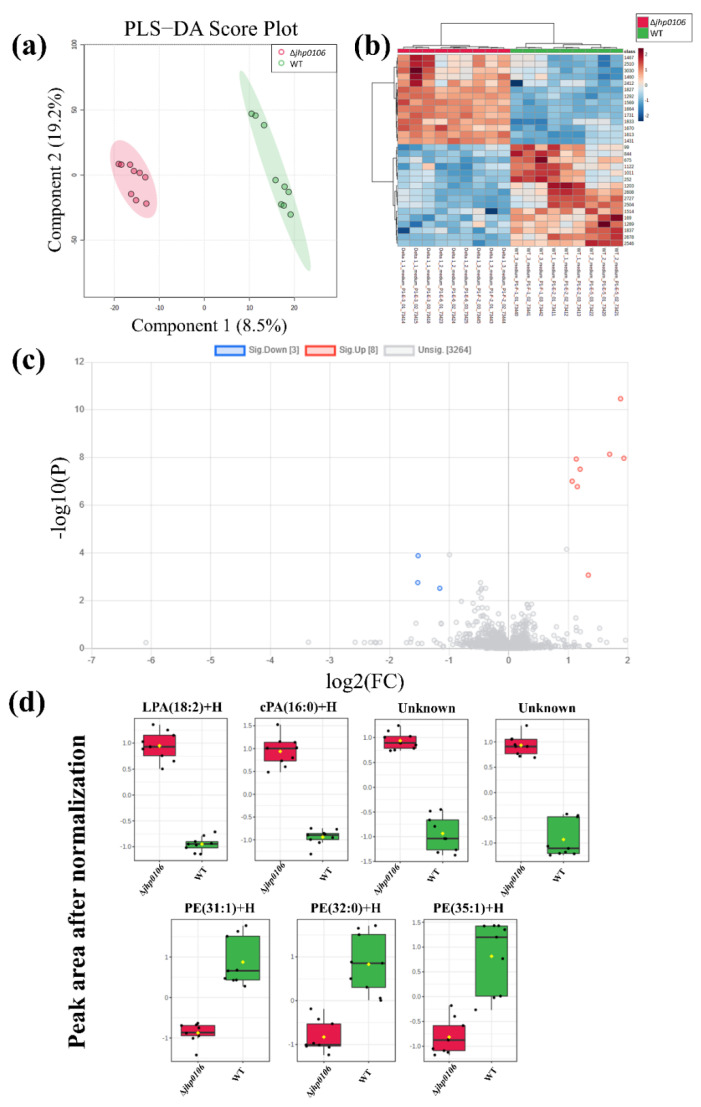
PLS−DA plot of the features obtained from the culture media of WT J99 and Δ*jhp0106 H. pylori* (**a**); heat map revealing features with higher VIP scores from the PLS−DA model (**b**); volcano plot (**c**) and boxplots (**d**) of features with a fold change of >2 and a *p* value of <0.05. The *y*-axis of (**d**) represents the normalized peak area after the autoscaling normalization. (The black dots in (**d**) are the sample values; while the yellow dots are the mean values).

**Figure 3 metabolites-11-00808-f003:**
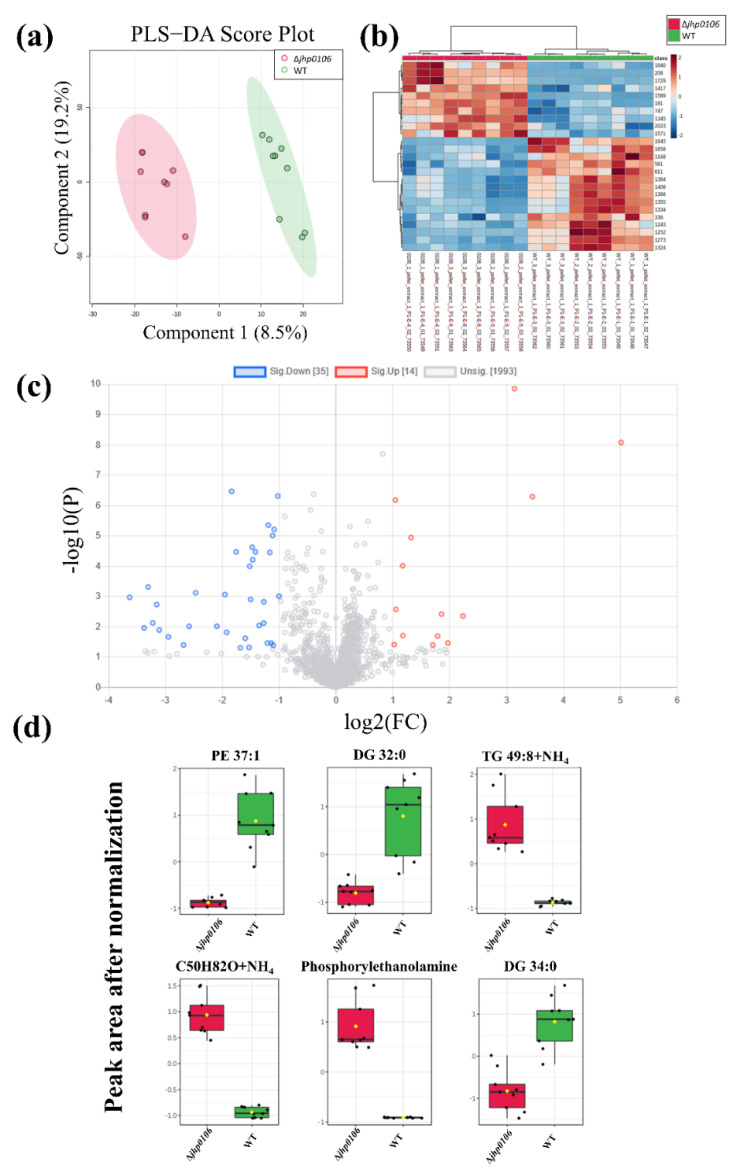
PLS−DA plot of the features obtained from WT J99 and Δ*jhp0106 H. pylori* pellets (**a**); heat map revealing features with higher VIP scores from the PLS−DA model (**b**); volcano plot (**c**) and boxplots (**d**) of features with a fold change of >2 and a *p* value of <0.05. The *y*-axis of (**d**) represents the normalized peak area after the autoscaling normalization. (The black dots in (**d**) are the sample values; while the yellow dots are the mean values).

**Figure 4 metabolites-11-00808-f004:**
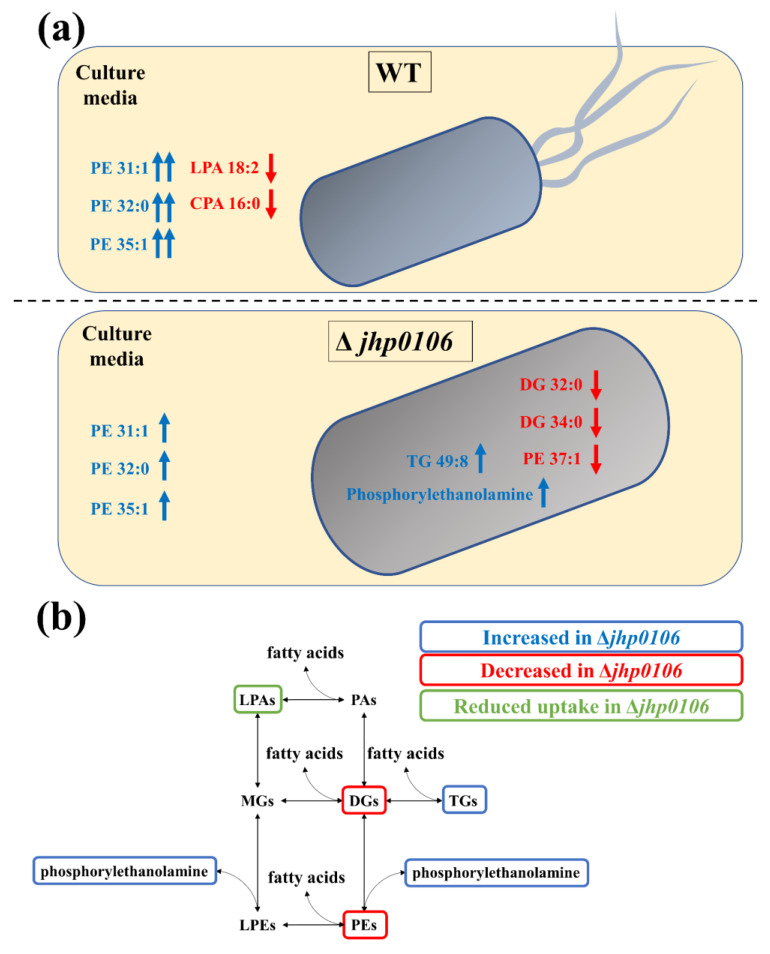
Scheme presenting tentatively identified metabolites in the culture media and within *H. pylori* (**a**); the illustrated pathways are associated with the tentatively identified metabolites (**b**).

## Data Availability

The data presented in this study are available from the corresponding author on request.

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
