# Peer review of "Untargeted Microbial Exometabolomics and Metabolomics Analysis of Helicobacter pylori J99 and jhp0106 Mutant"

_metabolites, 2021, doi:10.3390/metabo11120808_

Round 1

Reviewer 1 Report

The manuscript read well.

Minor points:

The first couple of sentences of the introduction are unnecessary since this journal is called "Metabolites" and mostly focused on the application of metabolomics. You can start by better defining exometabolomics.

The paragraph on untargeted vs targeted metabolomics can be deleted or significantly shortened.

The y-axis in figures 2D and 3D reads "normalized peak area" and the legend reads "peak area after autoscaling normalization". This is not sufficient. If the peak areas were normalized to a labeled internal standard, as recommended, then this should be stated clearly. If this was not performed then please describe what the y-axis actually presents.

Author Response

Minor points:

The first couple of sentences of the introduction are unnecessary since this journal is called "Metabolites" and mostly focused on the application of metabolomics. You can start by better defining exometabolomics.

Response:

        Thank you for your valuable suggestion. We shortened the introduction of metabolomics.

The paragraph on untargeted vs targeted metabolomics can be deleted or significantly shortened.

Response:

We appreciate your recommendation. We have removed the comparison of untargeted and targeted metabolomics.

The y-axis in figures 2D and 3D reads "normalized peak area" and the legend reads "peak area after autoscaling normalization". This is not sufficient. If the peak areas were normalized to a labeled internal standard, as recommended, then this should be stated clearly. If this was not performed then please describe what the y-axis actually presents.

Response:

        Thank you for your kind question. Actually, only the peak areas of targeted metabolites in figure 1 were normalized to the peak areas of ISs. For untargeted metabolic profiling, it is difficult to select suitable IS in advance for the identified unknown features. The MetaboAnalyst 5.0 (https://www.metaboanalyst.ca/) pipeline suggests standardizing the peak area of all the features by autoscaling mode (mean-centered and divided by the standard deviation of each metabolite) before the following multivariate analysis. We modified the y-axis labels in figures 2D and 3D as "peak area after normalization".

Reviewer 2 Report

The Manuscript from Kao et al. describes an untargeted metabolomics study looking at the exo and the endo metabolomics of helicobacter pylori and if differences in metabolomics pathways could be associated with the flagellar formation.

The methodology and the approach is valid and it could be easily used for other similar studies.

I have some comments regarding missing details in the method section and some conclusions.

1) How many samples and replicates were used in the study?

2) How the IS correction was done? It says that the matrix effect was corrected using the IS with similar RT. What does that mean in practice? Or what compound got corrected with what? Was it only the RT taking into account? How about the nature of the compound? 

3) It was used a human database for the compound ID, which is very odd considering that you have bacteria. Are the lipids you found plausible? 

4) Figure 2d and 3d. The LPA(18:2), the DG32:0 and the DG34:0 is represented 3 times, I would keep only one plot (the protonated version, for instance) , even though is among the selected features. I would actually do a dereplication step, before doing the data analysis.

5) The authors conclude that the extracted procedure could serve as a reference for future studies. I would consider rephrase it, simply because only one extraction procedure was tested. Furthermore it seems that methodology applied from sample prep to chromatographic analysis is oriented to more a lipid coverage. 

6) Title is not clear and redundant

Author Response

The Manuscript from Kao et al. describes an untargeted metabolomics study looking at the exo and the endo metabolomics of helicobacter pylori and if differences in metabolomics pathways could be associated with the flagellar formation.

The methodology and the approach is valid and it could be easily used for other similar studies.

I have some comments regarding missing details in the method section and some conclusions.

1) How many samples and replicates were used in the study?

Response:

        Thank you for your question. We performed 3 batch cultivation, and 3 technical replicates for every sample. We added the description in Section 2.2.

2) How the IS correction was done? It says that the matrix effect was corrected using the IS with similar RT. What does that mean in practice? Or what compound got corrected with what? Was it only the RT taking into account? How about the nature of the compound?

Response:

        We appreciate your constructive suggestion. We have purchased 5 SIL-ISs and 2 lipids (previously checked without interference in these samples) as the internal standards. In general, some of the demonstrated metabolites shown in figure 1 were corrected by their SIL-ISs or the ISs with similar RT, the lysoPCs and PCs were corrected by LysoPC 18:1 (IS), the PEs were corrected by PE 15:0/18:1 (IS), and the rest of metabolites were corrected by the ISs with similar RT. For this reason, we do consider the compound nature and similar retention times for the metabolites shown in figure 1. We have revised the description as “The matrix effect was corrected by using SIL-ISs or ISs considering the compound nature or with similar retention times (Table S2)”.

3) It was used a human database for the compound ID, which is very odd considering that you have bacteria. Are the lipids you found plausible?

Response:

        We appreciate your valuable question. We identified the tentative compounds mainly by the METLIN online database which consists of about 960,000 chemicals, and some of the metabolites were double confirmed by Human Metabolome Database. After identifying the tentative compound names, we further compared the MS2 fragments of tentative metabolites (if available) with reference standards to increase the reliability of the tentatively identified metabolites.

4) Figure 2d and 3d. The LPA(18:2), the DG32:0 and the DG34:0 is represented 3 times, I would keep only one plot (the protonated version, for instance) , even though is among the selected features. I would actually do a dereplication step, before doing the data analysis.

Response:

        Thank you for your kind suggestion. We removed the repeat metabolite features in figure 2d and 3d.

5) The authors conclude that the extracted procedure could serve as a reference for future studies. I would consider rephrase it, simply because only one extraction procedure was tested. Furthermore it seems that methodology applied from sample prep to chromatographic analysis is oriented to more a lipid coverage.

Response:

        We appreciate your constructive suggestion. We revised the sentence as “This study successfully demonstrates the feasibility of proposed extraction procedure and untargeted exometabolomic and metabolomic profiling strategy for microbial metabolomics.”

6) Title is not clear and redundant

Response:

        Thank you for your kind suggestion. We modified the title as “Untargeted Microbial Exometabolomics and Metabolomics Analysis of Helicobacter pylori J99 and jhp0106 Mutant”.

This manuscript is a resubmission of an earlier submission. The following is a list of the peer review reports and author responses from that submission.

Round 1

Reviewer 1 Report

In the manuscript entitled ‘Untargeted Microbial Exometabolomics and Metabolomics Reveal Differential Metabolic Profiles in Helicobacter pylori J99 and jhp0106 Mutant,’ the authors conducted untargeted metabolomics analysis on the exometabolome and metabolome of Helicobacter pylori with. The authors used various statistical analysis approaches to reveal metabolic features of relevance to phosphatidylethanolamine synthesis and flagellar formation.

Comments:

  1. Line 179. Do the authors mean ‘demonstrating significance?’
  2. Bacteria names should be in italics.
  3. The manuscript would benefit from having a flow chart of the data analysis workflow as a separate figure.
  4. Line 210. Do the authors have an explanation for why they observed elevated PE levels in the culture media? Also, would it be possible for the authors to plot intra- and extracellular levels of PE? It is not possible to compare their levels in figure 1.
  5. Lines 226-228. Not sure what the authors are trying to say regarding LPA uptake in this context. Can the authors clarify this?

Author Response

We are grateful for your constructive comments. Point-by-point responses to the raised comments are listed in the attached file.

Reviewer 2 Report

The first couple of paragraphs of the introduction are a bit long and partially irrelevant to the main focus of the study. Instead please expand the last 2 paragraphs of the intro and bring the focus back to why you chose to this study. 

Define exometabolomics. Define flagellin and its importance.

The results contain a significant amount of methods. The results section needs to focus mainly on the actual results not the methods. 

Figure 1 needs a better description. The current one does not include why the graphs were created, what the axes represent, etc.

What do you mean by "retention time alignment of 0.2min"? Is this an error window? 12 seconds seems too large of an error even for LC.

How many samples per subgroups, WT and MT, did you have?

Did you perform any multiple hypothesis testing?

There are some redundancies in the text (lines 175-193). 

What is the y-axis in Figures 2d and 3d? The figure legends need to be more descriptive.

Please provide MS/MS evidence to support the chemical IDs of all the metabolites mentioned in the paper (DG, LPA, PA, etc).

Author Response

(The authors gave the same response as above.)

Round 2

Reviewer 2 Report

Thank you for providing responses to the review.

The results section still contains a lot of information that belongs in the methods section. 

With n=3, there is no stat power to perform untargeted analysis. Your n should increase and multiple hypothesis testing needs to be applied.

To keep with the standards of practice in the field of metabolomics the "putative metabolites" highlighted in the manuscript should be validated against their respective reference chemical and the spectra data should be provided for review.

Author Response

The results section still contains a lot of information that belongs in the methods section. 

Response:

        Thank you for your advice. We have moved several paragraphs of bacteria sample preparation and the settings for nontargeted metabolic profiling into the methods section.

With n=3, there is no stat power to perform untargeted analysis. Your n should increase and multiple hypothesis testing needs to be applied.

Response:

        We appreciate your kind suggestions. We agree statistical power is not acceptable with n=3 for animal and clinical studies. For observation of metabolite alterations in the cultured pathogens or bacteria, most metabolomics studies include three or four biological replicates of each experimental treatment. (Dujardin et al., LC-MS metabolomics from study design to data-analysis – using a versatile pathogen as a test case, Comput Struct Biotechnol J. 2013; 4: e201301002.) In this study, we mainly focused on a unique strain of H. pylori with a specific gene mutation. To reduce the observation bias from the possible biological and analytical experiment, we performed 3 biological replicates (cultured in different batches) and 3 analytical technical repeats, in a total number of n=9 analyses in this study.

We are sorry for the confusing descriptions of multiple hypothesis testing used in this study. We used the web-based metabolomics data processing tool MetaboAnalyst 5.0 (https://www.metaboanalyst.ca/) to perform the statistics, and the p values shown in Fig. S1 are FDR corrected p values. We revised the descriptions in the supporting information and manuscript.

To keep with the standards of practice in the field of metabolomics the "putative metabolites" highlighted in the manuscript should be validated against their respective reference chemical and the spectra data should be provided for review.

Response:

        Thank you for your constructive suggestions. The use of reference standards is helpful for chemical identification, however, some of the identified compounds are not commercially available. Among these tentatively identified compounds, phosphorylethanolamine, PE(37:1), and TG(49:8) have lower abundance which hindered the obtainment of the MS2 spectrum. For this reason, according to the identified features we purchased three reference standards (cPA 16:0, DG 16:0/16:0, and PE 16:0/16:0) to confirm the tentatively identified compounds by using the retention time, accurate m/z, and MS2 spectrum from these reference standards.

        The reference standards showed the same retention time and accurate m/z as the tentatively identified compounds of cPA 16:0, DG 32:0, and PE 32:0. The MS2 spectrum of the reference standards (cPA 16:0, DG 16:0/16:0, and PE 16:0/16:0) and the tentatively identified compounds were provided in Fig S3 to Fig S8. Among these tentatively identified compounds, cPAs are cyclic LPAs. For identification of LPAs and cPAs, the fragment of m/z 153 is a common fragment spectrum from LPAs and CPAs, and sometimes the fragment of fatty acids could be also observed in the MS2 of LPAs and CPAs in the negative mode. For identification of PEs, the fragment of neutral loss of m/z 141 for PEs could be observed in the positive mode. For the identification of DGs, the fragments of fatty acid side chains could be observed in the MS2 spectrum.

        We have added the descriptions and the MS2 spectrums in the supporting information.
